# Hear you are: Teaching LLMs Spatial Reasoning with Vision and Spatial Sound

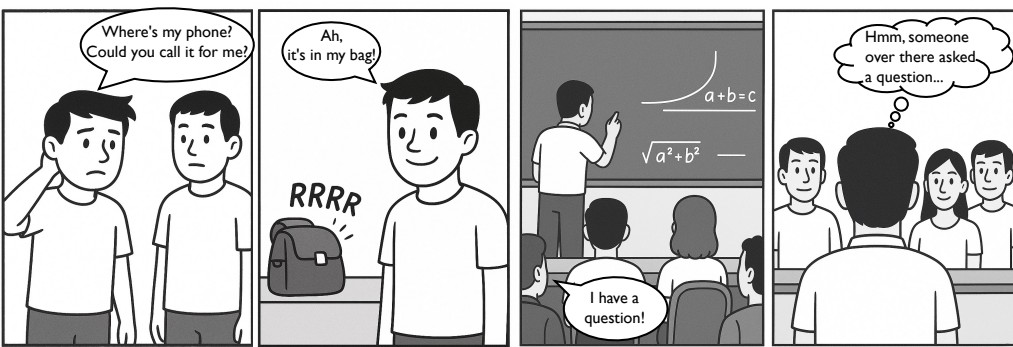

Figure 1: **Audio-Visual Spatial Reasoning.** (Left) A phone rings out of sight inside a bag; although the sound's semantic cue ("ring tone") is present, spatial reasoning is required to locate the true source among visually silent objects. (Right) In a classroom, several students share the same semantic cue ("speech"), so the teacher must rely on spatial audio to identify which student asked the question. These examples illustrate that accurate audio-visual understanding demands not only semantic alignment but also spatial comprehension.

## Abstract

Many audio-visual learning methods have focused on aligning audio and visual information, either through semantic or temporal correspondence. However, most of these works have utilized monaural audio, which does not contain information about the spatial location of the sound source. In contrast, humans and other animals utilize binaural hearing to perceive this spatial information. Combining spatial sound and visual perception enables powerful high-level reasoning: for example, a person looking for their phone may hear the ringing sound coming from a backpack sitting on a table, and quickly infer that the missing phone is inside the backpack. In this paper, we investigate the problem of **Audio-Visual Spatial Reasoning**. We design a spatial audio-visual question answering dataset to cover scenarios where semantic correspondence between audio and visual signals is absent but spatial alignment exists, as well as cases with multiple audio-visual semantic correspondences that require spatial reasoning to disambiguate. We propose a model that learns spatial comprehension across the audio and vision modalities by connecting them with a large language model and experimentally demonstrate that spatial sound perception is an essential part of our task.

# 1   Introduction

We live in a world full of sights and sounds, naturally associating what we hear with what we see. Several cues help us connect the two, such as the visual appearance and audible characteristics of an object, the synchronization between an action or event and its corresponding sound, and the direction from which the sound arrives, through binaural hearing. We rely on these audio-visual cues to locate a missing mobile device, or to know when an emergency vehicle is approaching as we are driving. This natural ability to connect auditory and visual information has motivated advancements in audio-visual machine learning, such as sound source localization (object detection based on audio queries) [7, 21, 33, 27, 30, 23, 29], source separation [3, 14, 16, 48, 46, 15, 45], and audio-visual synchronization [12, 6, 32]. However, most of these studies, which commonly use monaural audio, focus on the semantic correspondence between a sound and the visual appearance of the object that made the sound, or the audio-visual temporal alignment between an event and the sound it creates. These past approaches often overlook spatial cues that provide information about where a sound is coming from.

Binaural audio becomes essential when semantic matching is ambiguous or misleading. Figure 1 illustrates two scenarios where spatial reasoning is necessary. For instance, understanding that a ringtone sound is emanating from a backpack requires spatial reasoning, as the backpack does not semantically match the sound. Another example is when a single sound (e.g., speech) could correspond to multiple visual objects (e.g., several students in a classroom), where spatial cues help pinpoint the actual source. These examples highlight the limitations of previous methods, emphasizing the need to address spatial reasoning beyond basic perception.

Previous studies in spatial audio reasoning have primarily focused on audio-only approaches, excluding visual information while incorporating language as a modality for spatial interpretation. [13] aligns audio and text embeddings for spatial tasks, while [52] leverages large language models for spatial audio question answering. While spatial audio itself provides rich information for spatial reasoning, integrating visual information into these tasks is a natural extension, as visual signals inherently convey spatial context. This combination not only enhances spatial perception and localization capabilities, but also enables more sophisticated spatial reasoning, such as handling scenarios involving sounding sources and nearby visual objects.

In this paper, we address the problem of **Audio-Visual Spatial Reasoning**, which involves understanding the spatial relationship between a sound and the visual context. This task goes beyond simply perceiving and localizing a sound source, as it requires reasoning about spatial cues to infer relationships and interactions between objects. To support research on this problem, we construct a large-scale dataset of 1 million question-answer pairs, specifically designed to serve as both the training and evaluation set for spatial audio-visual reasoning in diverse scenarios. The vision and spatial audio is rendered using SoundSpaces 2.0 [4], with source audio clips sampled from VGGSound[8]. 3D objects associated with these sounds are generated using Stable Diffusion 3[35] and InstantMesh [49], and then are placed within the virtual environments. This dataset serves as a comprehensive benchmark for spatially intricate settings, providing questions that assess spatial alignment between modalities, relative locations between sounding and non-sounding objects, and localization of sound sources among multiple visual objects of the same category as the query audio.

Furthermore, we propose a multi-modal framework, Hear You Are LLM, which leverages spatial audio and visual encoders to integrate spatial information. The model is trained to handle all the spatial reasoning tasks from our dataset, enabling it to address scenarios where semantic alignment alone is insufficient. We experimentally demonstrate that our proposed method effectively addresses the audio-visual spatial reasoning problem, outperforming existing baseline models including a state-of-the-art monaural sound source localization method [39, 40] and a large language model-based audio-visual model that lacks spatial understanding. These results highlight the importance of incorporating spatial audio-visual knowledge to achieve robust multi-modal reasoning. To summarize, our main contributions are as follows:

- We define a new task, audio-visual spatial reasoning, focusing on understanding spatial relationships between sound and visual context, going beyond basic semantic perception such as sound source localization (object detection based on audio queries) and audio-visual segmentation.

- We propose *Hear You Are LLM*, a multi-modal modeling framework that integrates spatial audio and visual encoders with a large language model to handle complex spatial reasoning tasks.

- • We construct *Hear You Are QA*, the first large-scale dataset specifically designed for audio-visual spatial reasoning, consisting of 1 million question-answer pairs across diverse spatial scenarios for training and evaluation. We will open source both the dataset and the training code.

## 2 Related Work

### 2.1 Audio-Visual Sound Source Localization

Audio-visual sound source localization is the task of detecting the object or area that corresponds to the query audio in the visual scene. Following the development of deep learning, Senocak et al. [37, 38] suggested a semantic alignment-based approach by proposing a cross-modal attention mechanism with contrastive learning. The field has advanced in the direction of better cross-modal alignment by leveraging negative-free self-supervised learning [42], intra-modality similarity learning [43], and the use of multiple positive learning [39], aligning with representation learning methods. However, these methods rely on monaural audio and are limited to audio-visual semantic correspondence without spatial understanding.

Different approaches have focused more on spatial audio for sound source localization. Anoopcherian et al. [20] proposed a 3D sound source localization method trained on a dataset with four-channel audio and multi-view visual scenes synthesized using SoundSpaces 2.0. Their approach localizes sound within the visual scene, but the visual counterpart of the sound is not visible in their setting, as they only localize the area of the sound source. Shimada et al. [41] constructed an audio-visual sound source localization and detection dataset in which audio-visual alignment is guaranteed. In their framework, the visual signal serves as an auxiliary modality to improve sound localization and detection. In contrast, we present an audio-visual scene that includes both sound-producing and silent objects, allowing the model to learn a broader range of spatial reasoning tasks that require contextual understanding beyond basic localization.

### 2.2 Spatial Audio Reasoning

Following recent advancements in audio understanding [18, 1, 24] and reasoning [19, 36], several approaches have been proposed to address spatial audio reasoning. [52] synthesize the spatial sound question answering dataset with the SoundSpaces 2.0 simulator and train a spatial audio encoder and a large language model for spatial audio understanding and reasoning. This framework handles tasks such as sound event detection, direction and distance estimation, and spatial reasoning, for example, "What is the sound on the left side of the sound of the dog barking?" Another line of research explores spatial audio reasoning through contrastive language-audio pretraining, with synthetic first-order ambisonics [13]. However, these approaches do not incorporate the vision modality, which opens another dimension for reasoning.

### 2.3 Audio-Visual LLMs

Inspired by the advancements of Large Language Models (LLMs), recent studies have extended these models to Multimodal Large Language Models (MLLMs) to tackle a wider range of multimodal tasks. In the audio-visual domain, GroundingGPT [26] introduces multimodal grounding for audio, image, and video data using LLMs. Meerkat [10] aligns audio-visual features using optimal transport and attention consistency, and CAT [51] aggregates question-related clues in audio-visual scenarios. From a benchmarking standpoint, AVHBench, AVTRUSTBENCH, and AV-Odyssey Bench [11, 44, 17] provide comprehensive benchmarks targeting hallucination detection [44], reliability and robustness [11], and both foundational capabilities and high-level reasoning [17]. While recent studies have advanced multimodal learning, they primarily rely on monaural audio, limiting their ability to handle spatial reasoning. As spatial reasoning enables a broader range of tasks and more closely reflects real-world scenarios, it must be addressed to achieve comprehensive audio-visual understanding. We propose a new dataset and model specifically designed for spatial reasoning in audio-visual tasks.

## 3 Creation of Hear You Are QA Dataset

Our goal is to train a model to learn both semantic and spatial reasoning, for audio-visual inputs. To this end, we introduce the Hear You Are QA Dataset. Constructing large-scale audio-visual scene data

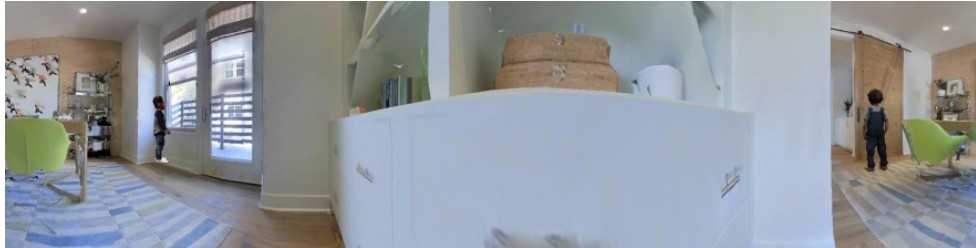

Figure 2: **Image sample from Hear You Are QA dataset.** The dataset consists of diverse indoor scenes captured in 360° panoramic views, featuring various object arrangements and providing a comprehensive range of spatial contexts for analysis.

Table 1: Spatial audio visual question types and base prompts/answers

| |
|---|
| **Q1. Spatial Correspondence** |
| **Q:** What is the sound class category? Where is the sound coming from? **A:** phone ringing; cupboard |
| **Q2-4. Relative Location** |
| **(Distance) Q:** Is the sound source of the siren closer to the agent than it is to the cat? **A:** Yes |
| **(Direction) Q:** Can you estimate the distance from the accordion sound to the dog, and the relative location of the accordion from the dog? **A:** right; behind; upper; 2.3 m |
| **(Angle) Q:** Can you estimate the distance from the accordion sound to the dog, and the angle between the agent's gaze directions toward the accordion and the dog? **A:** 30; 10; 2.3 m |
| **Q5. Spatial & Semantic Correspondence (One visual object semantically matches the audio)** |
| **Q:** What is the object in the scene located at $(-30, -12)$, 2.549 m? Is it making a sound? **A:** bird squawking; making sound |
| **Q6. Spatial & Semantic Correspondence (Multiple visual objects semantically match the audio)** |
| **Q:** What is the object in the scene located at $(150, -14)$, 1.735 m? Is it making a sound? **A:** canary calling; making sound |
| **Q7. Spatial & Semantic Correspondence (One visual object semantically matches the audio)** |
| **Q:** Given multiple visual objects, which one is making a sound, and where is it located? **A:** bird squawking; $-30$; $-12$; 2.549 m |
| **Q8. Spatial & Semantic Correspondence (Multiple visual objects semantically match the audio)** |
| **Q:** Could you determine the sound class category, and which object of that category in the scene is making the sound? **A:** canary calling; 150; $-14$; 1.735 m |
| **Q9. Semantic Co-occurrence** |
| **Q:** What is the sound class category? Is the sound source visible in the scene? **A:** cat; not visible |
| **Q:** What is the sound class category? Is the sound source visible in the scene? **A:** fox; visible |

with real-world spatial audio is time-consuming and challenging, requiring specialized equipment such as ambisonic or dummy head microphones. To efficiently build a diverse dataset with various objects and sound events, we adopt a simulation-based approach to generate both the scenes and spatial audio.

**Spatial Audio Simulator.** We employ the SoundSpaces 2.0 simulator [4], which renders geometry-based acoustics, adding realistic reverberation for any source–receiver pair. Users can freely vary wall materials, object properties, and microphone-array geometry, letting us create a rich, controllable dataset while retaining exact ground-truth parameters, e.g., every source's 3D position and orientation. Scene meshes come from Matterport3D [2], a collection of 90 fully scanned buildings averaging 24.5 rooms across 2.61 floors and 517.34 m² of floor space. We use 72 scenes for training, 9 for validation and 9 for testing. Given a source location, monaural signal, receiver position, and heading, the observed signal is obtained by convolving the monaural signal with the environment's room impulse response. We configure the receiver to record a binaural audio signal with the default Head Related Transfer Function (HRTF) provided by SoundSpaces2.0.

**Sound Sources.** Previous spatial audio datasets include either a limited number of class categories [41] or classes that are not guaranteed to be visually observable [52, 20]. To construct a large-scale audio-visual dataset, we adopt VGGSound [8], which contains 200,000 in-the-wild 10-second YouTube clips, each annotated with one of 309 audio event classes. However, some of these classes correspond to events that typically occur outdoors or are difficult to associate with a single visual object (e.g., "Airplane Flyby", "People Marching"). To enhance the visual reliability and realism of our dataset, we manually exclude categories typically occur outdoors, or are visually

ambiguous. We follow the original testing splits provided by VGGSound, and create a validation set of the same size as the testing set by sampling clips from the VGGSound training split.

**Visual Objects.** Due to the limited number of sound-emitting categories in existing 3D object datasets, we generate our own 3D objects to be placed within the Matterport3D environments, either as sounding objects or as distractor objects. Specifically, we first select 150 class categories from VGGSound and 40 from ImageNet, and generate 2D images for each category using Stable Diffusion 3. After manually filtering out low-quality or unrealistic generations, we select 40 visually plausible images per category. These 2D images are then lifted into 3D object meshes using the method from [49]. For each sounding object category, we reserve 32 images for training, 4 for validation and 4 for testing.

**Audio-Visual Scene Construction.** Each audio-visual scene consists of a 360° panoramic image as Figure 2 and corresponding binaural audio. We stitch 18 images, each with a horizontal FoV of 20 degrees as in [5], to form a 360° view. The final image resolution is set to 224×812, and the center of the image is aligned with the front-facing direction of the observing agent in SoundSpaces 2.0.

We inject the aforementioned sound source and 3D objects into random locations within the scene, excluding placements where objects are occluded by walls or located in a different room. Each scene includes one sound source. The sound source, depending on the question scenario, is assigned to either a semantically matching object from a VGGSound category, a random object from a different category (VGGSound or ImageNet), or a random empty location within the scene.

One potential concern is that rendering artifacts, such as visible seams between injected objects and the original scene, could serve as shortcuts for the model. To mitigate this and increase the visual complexity of the scene, we randomly insert up to three random objects sampled from categories distinct from the main visual objects in the scene.

**Crafting Questions.** We manually defined nine different "base" questions that require spatial audio-visual understanding, summarized in Table 1. When filling a question template, we use handcrafted rules to automatically populate the missing fields in the question and answer using the scene construction parameters. The questions cover four main categories: spatial correspondence (Q1), relative location (Q2, Q3, Q4), spatial and semantic correspondence (Q5, Q6, Q7, Q8), and semantic co-occurrence (Q9). **Spatial Correspondence** questions aim to evaluate whether the model can correctly associate an audio signal with its spatially aligned visual source. To assess the model's robustness, we include counterfactual examples in which semantically mismatched visual objects and sounds (e.g., a piano and dog barking) are placed at the same location. This setting discourages reliance on semantic priors and encourages the model to learn true spatial correspondence between audio and visual modalities without hallucination. **Relative Location** questions assess the model's ability to understand the spatial relationship between audio and visual information. These include determining whether a sound source is located to the left, right, front, or behind the agent, as well as reasoning about vertical position (e.g., above or below), angular direction, and relative distance with respect to a visual reference. **Spatial and Semantic Correspondence** questions evaluate whether the model can jointly associate the correct object class (semantic) and its location (spatial) based on the audio signal. **Semantic Co-occurrence** questions focus on learning spatial audio understanding regardless of whether the corresponding visual object is explicitly visible, encouraging the model not to solely rely on an object's appearance. To diversify the question set and improve naturalness, we utilize ChatGPT-4o to paraphrase and expand each base question into multiple human-like variations.

# 4  Method

Our aim is to construct a model that can answer the questions in our proposed dataset by leveraging both visual and spatial audio inputs. To this end, we design and train a multi-modal large language model with both visual and binaural audio inputs. The overall architecture is illustrated in Figure 3.

**Audio and Visual Encoders with Projector.** Given an image $v$ and its corresponding audio $a$, our backbone networks extract features from each modality. The vision encoder $f_v$ processes a panoramic image frame and outputs a sequence of spatially aligned visual tokens, $\mathbf{v} \in \mathbb{R}^{N_v \times C_v}$, where $N_v$ is the number of visual tokens and $C_v$ is the feature dimension of each token. We preserve the full spatial layout of patch tokens without pooling. The audio encoder $f_a$ takes the input spectrogram of $a$ and produces a set of audio tokens, $\mathbf{a} \in \mathbb{R}^{N_a \times C_a}$, where $N_a$ is the number of audio tokens and $C_a$ is the corresponding feature dimension. Each modality-specific encoder is followed by a

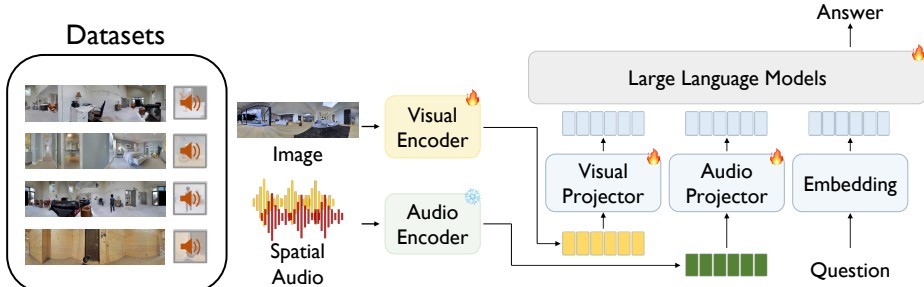

Figure 3: **The pipeline of our framework:** feature extraction, projection, and multimodal reasoning. We extract spatial audio and visual features using pre-trained encoders, project them into a shared embedding space, and integrate the embeddings with the question embedding to generate the answer.

projector that maps the extracted features into the hidden dimension of the language model. The visual projector attends to the spatial visual features to generate $N_V$ projected tokens, and the audio projector similarly produces $N_A$ tokens from the audio features. These projected tokens are then passed to the large language model for multi-modal reasoning.

**Large Language Model.** To bridge the audio and visual encoders, we utilize a large language model that takes as input the projected audio and image tokens along with the embedded question text. During fine-tuning, the model is optimized to generate the correct answer based on the given question and the corresponding multimodal inputs. Training is performed using the standard language modeling objective function that maximizes the likelihood of the target sequence using a cross-entropy loss applied at each token position.

**Warm Start of the Encoders.** To ensure the effectiveness of each modality-specific representation, the audio and visual encoders, along with their respective projectors, are pretrained in a unimodal setting using a large language model. We utilize the panorama image and binaural audio from our dataset and construct two types of auxiliary questions for each modality: classification and localization tasks. For the visual encoder, the classification task involves identifying visual objects at specific coordinates, phrased as "What visual objects did you detect at ({azimuth}, {elevation}), {distance} meters?", and the localization task asks for the predicted azimuth, elevation, and distance to a specified object class, stated as "What are the predicted azimuth and elevation angles, and the distance to the {class category}?". The audio encoder is trained with analogous tasks: the classification task asks "What sound did you detect?", while the localization task prompts for spatial coordinates of the sound source with the question "What are the predicted azimuth and elevation angles, and the distance to the sound source?". The visual encoder adopts a progressive training scheme, first focusing on classification to learn semantic representations and then incorporating spatial grounding through a combined classification and localization task. The audio encoder is trained on both tasks jointly from the beginning.

## 5 Experiments

### 5.1 Implementation Details

**Image Encoder** $f_v$**.** We use a SigLIP2 [47] vision encoder with the NaFLEX setting, which supports flexible image resolutions and aspect ratios. The encoder processes a panoramic image and outputs a sequence of patch tokens. We apply LoRA [22] to fine-tune the patch embedding and attention layers of the encoder during both the uni-modal training and the audio-visual end-to-end training.

**Audio Encoder** $f_a$**.** We use the pretrained Spatial-AST binaural audio encoder from [52]. The model takes binaural audio spectrograms as input and generates a sequence of audio tokens that preserve spatial acoustic cues. The encoder was pretrained using the same audio event classification and localization tasks proposed in [52]. This encoder is kept frozen throughout the entire training process.

**Modality-specific Projectors and Large Language Model** We adopt the Q-Former architecture as the projector for both modalities. The audio-side projector is based on the implementation and pretrained weights from BAT [52], while the visual-side projector is adapted from BLIP-2 [25], using only the first two attention layers and their corresponding pretrained weights. The number of query

Table 2: **Evaluation of baseline models on sound source localization that requires spatial understanding.** R, B, M, Q refer to RGB Image, Binaural Audio, Monaural Audio, and Question (Text) in this table.

| Method | Modality | Q1 (class) | Q1 (aligned) | Q1 (non-matching) | Q7 (class) | Q7 (DoA) | Q8 (class) | Q8 (DoA) |
|---|---|---|---|---|---|---|---|---|
| Question Only | Q | 3.50 | 3.00 | 2.44 | 2.56 | 7.89 | 0.78 | 7.61 |
| ISSL [39, 40] | R+M | 26.97 | 28.83 | 12.94 | 28.46 | 23.18 | 26.94 | 21.0 |
| ACL-SSL [31] | R+M | 40.56 | 32.83 | 10.61 | 40.41 | 30.68 | 41.11 | 24.33 |
| VideoLLaMA2 [9] | R+M+Q | 51.01 | 77.44 | 50.75 | 70.88 | 68.57 | **75.33** | 46.37 |
| Ours | R+B+Q | **52.69** | **77.61** | **61.67** | **75.44** | 73.21 | 70.27 | **64.27** |

237 tokens is set to $N_1 = 64$ for audio and $N_2 = 128$ for vision. All projector parameters are fully
238 trainable. We adopt Qwen2-7B-Instruct [50] as our LLM backbone.

239 **Training Setup and Input Preprocessing.** Inputs to our model consist of a single $224 \times 812$
240 panoramic image and a 10-second audio binaural waveform sampled at 32 kHz. We preprocess the
241 image input following [47] and the audio input following [52]. Our full model is trained for 3 epochs
242 on 8 A5000 GPUs with an effective batch size of 128, using a LoRA rank of 16 for the image encoder
243 and LLM backbone. The training takes three days. Additional training details are provided in the
244 supplementary material.

245 **Baselines.** Since no existing method directly addresses our proposed task, we introduce three baselines
246 adapted from related domains. The first two baselines are audio-visual sound source localization
247 approaches. Specifically, we adopt the framework proposed in [39, 40], which has demonstrated
248 strong performance on synthetic benchmarks and exhibits robustness with multiple visual objects.
249 [31] learns audio-driven embeddings compatible with the text encoder of CLIP[34] and leverages
250 the CLIP-based segmentation network [28] to achieve tight localization results. Although they do
251 not handle language understanding, we evaluate them using cross-modal retrieval and localization
252 metrics. Implementation details are provided in the supplementary material. The third baseline is
253 the VideoLLaMA2[9], multi-modal large language model (MLLM), the closest prior work to ours
254 in terms of multimodal reasoning. For a fair comparison, we replace its original vision and audio
255 encoders with the same encoders used in our method, Spatial AST[52] and SigLIP2 NaFLEX [47],
256 and fine-tune the model on our proposed dataset using the same LLM backbone. Notably, the baseline
257 uses monaural audio input, whereas our method leverages binaural cues. Since the sound source
258 localization approaches are not designed for reasoning tasks (e.g., Q2, Q3, Q4, Q5, Q6, Q9), we
259 evaluate them only on tasks that do not require language processing. The metrics in Table 2 cover
260 classification and direction of arrival (DoA). Q1 (aligned) and Q1 (non-matching) indicate sound
261 source localization task where the source is semantically aligned and non-aligned with the audio,
262 respectively.

263 ## 5.2 Main Results

264 We present our results in Table 2, showing that only our model effectively addresses spatial reasoning
265 scenarios. For sound classification tasks (Q1, Q7, Q8), sound source localization approaches
266 outperform the Question Only setting, which serves as a random baseline. VideoLLaMA2 shows
267 comparable performance to our model, particularly in Q1 (aligned) and Q7 (DoA), where semantic
268 cues are sufficient for localization due to the presence of a single matching visual object with audio.
269 Monaural audio is sufficient to localize the sound source, allowing baseline models to perform
270 consistently without spatial audio cues. However, in Q1 (non-matching) and Q8 (DoA), spatial
271 reasoning is essential for different reasons. In Q1 (non-matching), the visual object at the sound
272 source is semantically unrelated to the audio, requiring spatial cues to correctly associate the sound
273 with the aligned object. In Q8 (DoA), multiple objects share the same sound category, making it
274 necessary to differentiate between them using spatial cues. In both cases, baseline models perform
275 significantly worse. VideoLLaMA2, which shares the same architecture as ours but lacks binaural
276 audio, achieves approximately 50% accuracy in Q8 (DoA), indicating its inability to distinguish
277 between visually similar objects that semantically match the audio. Since all baseline models use
278 only monaural audio, they lack spatial information, making spatial reasoning impossible.

Table 3: **Ablation study on modality settings for audio-visual spatial reasoning tasks.** R, B, M, Q refer to RGB Image, Binaural Audio, Monaural Audio, and Question (Text) in this table.

| metric | Trained and tested on | | | | | Trained on R+B+Q, tested on | | Random Chance |
| | R+B+Q | R+M+Q | B+Q | M+Q | R+Q | R+M+Q | B+Q | Q |
|---|---|---|---|---|---|---|---|---|
| *Q1* | | | | | | | | |
| sound accuracy ↑ | 52.69 | 51.01 | 52.53 | 51.40 | 27.28 | **54.03** | 46.86 | 3.50 |
| coming-from accuracy ↑ | **69.64** | 64.10 | 26.40 | 26.40 | 56.22 | 61.92 | 23.39 | 2.72 |
| *Q2* (Yes or No) ↑ | 84.74 | 83.77 | 55.63 | 50.87 | **85.28** | 83.55 | 54.11 | 50.11 |
| *Q3* | | | | | | | | |
| 3-field accuracy ↑ | 69.73 | 66.52 | 32.40 | 18.67 | **74.46** | 66.42 | 24.57 | 18.56 |
| Avg. distance error (m) ↓ | 0.39 | 0.41 | 1.20 | 1.31 | **0.36** | 0.47 | 1.37 | 1.34 |
| *Q4* | | | | | | | | |
| DoA accuracy ↑ | **65.68** | 59.03 | 12.86 | 12.43 | 58.06 | 56.14 | 11.38 | 9.80 |
| Avg. DoA error (°) ↓ | **15.41** | 20.21 | 81.18 | 87.38 | 18.59 | 23.55 | 86.49 | 85.48 |
| Avg. distance error (m) ↓ | **0.38** | 0.47 | 1.10 | 1.21 | 0.38 | 0.51 | 1.32 | 1.21 |
| *Q2-invisible audio* ↑ | 72.46 | 70.40 | 57.14 | 48.00 | **73.03** | 70.51 | 52.91 | 50.63 |
| *Q3-invisible audio* | | | | | | | | |
| 3-field accuracy ↑ | **59.52** | 47.29 | 34.14 | 18.45 | 41.64 | 45.56 | 25.49 | 18.22 |
| Avg. distance error (m) ↓ | **0.75** | 0.98 | 1.20 | 1.33 | 1.02 | 1.12 | 1.39 | 1.38 |
| *Q4-invisible audio* | | | | | | | | |
| DoA accuracy ↑ | **41.18** | 16.71 | 11.18 | 11.76 | 13.53 | 16.47 | 11.29 | 9.88 |
| Avg. DoA error (°) ↓ | **39.81** | 69.25 | 80.51 | 84.56 | 77.15 | 75.39 | 85.24 | 84.81 |
| Avg. distance error (m) ↓ | **0.71** | 1.08 | 1.13 | 1.21 | 1.16 | 1.04 | 1.32 | 1.23 |
| *Q5* | | | | | | | | |
| class accuracy ↑ | 72.43 | 74.26 | 25.79 | 25.63 | **74.87** | 72.82 | 22.18 | 2.78 |
| sounding accuracy ↑ | 75.60 | 64.54 | 59.48 | 37.72 | 36.63 | 65.93 | **75.93** | 41.36 |
| *Q6* | | | | | | | | |
| class accuracy ↑ | **81.06** | 81.61 | 51.78 | 50.47 | 83.78 | 80.72 | 42.72 | 3.72 |
| sounding accuracy ↑ | 72.33 | 52.33 | 59.33 | 38.67 | 31.94 | 49.28 | **75.67** | 41.67 |
| *Q7* | | | | | | | | |
| class accuracy ↑ | **75.44** | 70.88 | 51.64 | 53.62 | 37.35 | 73.53 | 51.68 | 2.56 |
| DoA accuracy ↑ | **73.21** | 68.57 | 47.30 | 7.80 | 37.52 | 64.04 | 48.38 | 7.89 |
| Avg. DoA error (°) ↓ | **14.75** | 22.41 | 33.02 | 88.31 | 56.66 | 24.55 | 35.25 | 90.92 |
| Avg. distance error (m) ↓ | **0.30** | 0.33 | 0.50 | 0.53 | 0.44 | 0.36 | 0.79 | 0.53 |
| *Q8* | | | | | | | | |
| class accuracy ↑ | 70.27 | **75.33** | 48.42 | 48.02 | 69.89 | 71.90 | 32.51 | 0.78 |
| DoA accuracy ↑ | **64.27** | 46.37 | 47.69 | 8.46 | 43.72 | 39.76 | 49.41 | 7.61 |
| Avg. DoA error (°) ↓ | **23.78** | 50.80 | 32.32 | 89.93 | 51.90 | 52.46 | 32.45 | 89.40 |
| Avg. distance error (m) ↓ | **0.36** | 0.44 | 0.48 | 0.51 | 0.42 | 0.46 | 0.85 | 0.52 |
| *Q9* | | | | | | | | |
| sound accuracy ↑ | 54.00 | 51.14 | 51.14 | 52.20 | 27.17 | **55.57** | 47.25 | 2.81 |
| visiblity accuracy ↑ | 75.22 | 72.94 | 38.99 | 39.79 | 33.31 | **76.35** | 49.42 | 42.31 |

## 5.3 Ablation Studies

Table 3 shows that both image (R: RGB) and binaural audio (B) inputs are crucial for spatial reasoning. It compares R+B+Q, R+M+Q (M: monaural), B+Q, M+Q, and R+Q (Q: question), highlighting that binaural audio provides spatial cues while monaural lacks directional information. The following is an analysis of the performance for each question type.

**Question 1** involves sound and visual object classification, with half of the samples containing a non-matching visual object at the sound source. Both R+B+Q and R+M+Q show similar sound classification accuracy (52.69% and 51.01%), suggesting comparable semantic cues from monaural and binaural audio. However, in coming-from accuracy, R+B+Q (69.64%) outperforms R+M+Q (64.10%), highlighting the spatial advantage of binaural audio.

**Questions 2, 3, and 4** assess distance and relative location between the sound source and visual objects, requiring spatial reasoning across modalities. For visible audio, R+M+Q achieves 66.52% in Q3 and 59.03% in Q4, performing similarly to R+B+Q (69.73% and 65.68%). When the sound source is invisible, R+B+Q shows a clear advantage, outperforming R+M+Q in Q3 (59.52% vs. 47.29%) and Q4 (41.18% vs. 16.71%). This highlights the role of binaural audio in capturing spatial cues that monaural audio with visual input cannot provide.

**Questions 5 and 6** both involve identifying the sound-producing object but differ in complexity

based on the number of visual objects that match the sound. In Q5, with only one matching object, visual context alone provides sufficient spatial information for localization. R+M+Q leverages visual cues effectively, achieving a sounding accuracy of 64.54%. With no visual ambiguity, the model can reliably associate the sound with the correct object using spatial information from the visual signal. In Q6, two visually similar objects match the sound, introducing ambiguity. R+M+Q's performance drops to 52.33%, as visual context alone is no longer sufficient to distinguish between the two objects, leading to random guessing. In contrast, B+Q and R+B+Q maintain consistent performance across both questions. In Q5, they achieve 59.48% and 75.60%, respectively, and in Q6, their performance remains stable at 59.33% and 72.33%. This stability is due to binaural audio, which provides explicit spatial cues, enabling the model to localize the sound source based solely on directional information, unaffected by visual similarity. These results indicate that when there is only one matching object (Q5), R+M+Q can effectively use visual spatial information. However, when multiple visually similar objects are present (Q6), spatial audio cues become essential, allowing B+Q and R+B+Q to maintain stable performance regardless of visual similarity. These results highlight the importance of binaural audio in resolving ambiguity in complex visual scenes.

**Questions 7 and 8** both involve sound classification and localization but differ in the number of visual objects that correspond to the audio, with two in Q8 and one in Q7. In Q8, two visually similar objects correspond to the audio, making it difficult for the model to distinguish between them using visual information alone. R+M+Q and B+Q show similar DoA accuracy (46.37% and 47.69%), but their Avg. DoA errors differ, with R+M+Q at 50.80° and B+Q at 32.32°. R+M+Q relies on visual context for spatial cues, but semantic ambiguity between the two objects complicates localization, leading to random selection and higher error. In contrast, B+Q, using binaural audio, focuses solely on directional information, perceiving only one sound source without considering object-level ambiguity, resulting in a lower error. R+B+Q achieves the lowest error (23.78°) by combining spatial audio and visual inputs. In Q7, the audio corresponds to a single object, eliminating semantic ambiguity. In this case, the performance of R+M+Q and B+Q reverses from Q8. R+M+Q records a lower error (22.41°) than B+Q (33.02°), indicating that when only one object is present, visual spatial information can effectively guide localization without semantic confusion. These results support the findings in Q5 and Q6, emphasizing the role of spatial audio in disambiguating visually similar objects.

**Question 9** involves sound classification and localization while also requiring the model to determine whether the object is visually present at the sound source. This task demands both audio and visual semantic understanding. Both multi-modal settings (R+B+Q, R+M+Q) successfully address this question.

**Modality Setting Cross-Evaluation.** To assess the impact of vision signals and binaural audio during training, we evaluate the model trained on R+B+Q under R+M+Q and B+Q settings. While Q7 and Q8 show minimal change, Q5 and Q6 exhibit noticeable gaps in sounding accuracy. This might come from Q5 and Q6 only requiring yes/no responses given a location, without the detailed localization required in Q7 and Q8. Consequently, the model in the B+Q setting may not effectively leverage spatial reasoning for these tasks. However, with visual signals, the model gains implicit spatial cues that align audio locations with the visual scene, potentially enhancing spatial audio understanding. Thus, the presence of visual information may be beneficial even for learning spatial audio cues.

## 6 Conclusion

We introduce a new task, audio-visual spatial reasoning, along with the *Hear You Are LLM* and QA dataset. Unlike prior work that focuses on semantic or temporal alignment, our approach emphasizes spatial reasoning by integrating binaural audio and visual inputs. We build a large-scale dataset covering diverse spatial scenarios and propose a multimodal framework combining spatial encoders with a large language model. Experiments show that monaural audio with vision or unimodal binaural methods lack the capacity for spatial reasoning. These results underscore the importance of spatial reasoning in robust multimodal understanding and set a new benchmark in audio-visual learning.

## 7 Limitations and Future Directions

While our framework effectively addresses spatial reasoning by integrating binaural audio and visual context, several real-world scenarios remain unaddressed. Specifically, our approach does not consider moving sound sources, actions associated with visual objects, or occluded objects positioned behind walls or in separate rooms. These aspects are critical for capturing dynamic spatial interactions. Future work will focus on extending the dataset to incorporate these complexities, enabling more comprehensive audio-visual reasoning in realistic settings.

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
