# OpenReview forum: "Hear you are: Teaching LLMs Spatial Reasoning with Vision and Spatial Sound"
_NeurIPS.cc/2025/Conference — Submitted to NeurIPS 2025_

### Official Review · Reviewer_doKX · 2025-07-02

**Clarity:** 3
**Significance:** 3
**Originality:** 3
**Rating:** 4
**Confidence:** 4

**Summary:**

This paper introduces Audio-Visual Spatial Reasoning, a task requiring models to infer spatial relationships between binaural audio and panoramic images. To support it, the authors build Hear You Are QA, a large-scale simulated dataset of 1 M question–answer pairs covering spatial correspondence, relative location, semantic alignment, and co-occurrence scenarios. They propose Hear You Are LLM, which encodes binaural audio and 360° vision via pretrained encoders, projects them into a shared token space, and feeds them (with the question) into a frozen LLM fine-tuned end-to-end. Experiments show that only the full model (vision + binaural + text) consistently solves tasks requiring spatial cues, outperforming monaural or vision-only baselines.

**Questions:**

See Weaknesses.

**Ethical Concerns:**

["NO or VERY MINOR ethics concerns only"]

**Final Justification:**

The authors have addressed most of my concerns and included additional experiments, but I would like to keep my rating as Borderline Accept. What I really wish is that the author could provide real-world evaluations, which I think are crucial for evaluating generalization.

**Limitations:**

Yes.

**Quality:**

3

**Strengths And Weaknesses:**

Strengths:
1. Constructs a diverse QA dataset using SoundSpaces 2.0, VGGSound clips, and automatically generated 3D objects, enabling controlled evaluations across nine question types.
2. Compares against strong baselines: two audio-visual localization methods and a monaural-audio VideoLLaMA2 adaptation.
3. Ablation studies systematically vary modalities (R+B+Q, R+M+Q, B+Q, etc.) and visible vs. invisible sources.

Weaknesses:
1. All experiments use simulated scenes. It remains unclear how models transfer to real binaural recordings with natural reverberation and visual complexity. I think the paper would benefit from recording a small real-world test set (e.g., 360° video + binaural audio in indoor rooms) and reporting zero-shot performance or lightweight fine-tuning results to demonstrate generalizability.
2. Real scenes often have simultaneous sounds, but each scene of this dataset contains a single sound source, which simplifies the task. Please consider including experiments with two concurrent sound sources (e.g., speech + music) to test the model’s capacity for multi-source spatial reasoning or clarify why single-source suffices for the proposed task.
3. Although the paper mentions inserting distractor objects to mask seams, it does not quantify whether visual "stitching" artifacts could leak location. Could the authors analyze or ablate reliance on low-level cues (e.g., by blurring object boundaries) to ensure the model truly uses spatial audio rather than visual shortcuts?

---

> ### Author Rebuttal · Authors · 2025-07-30
>
> We sincerely appreciate the insightful comments on real-world scenarios, multi-source spatial reasoning, and stitching artifacts. In particular, the suggestion regarding multi-source settings inspired an interesting analysis that enriched our understanding of the model's behavior.
>
> **Real-world Scenario Discussion**
>
> Please read rebuttal to yWdi
>
> **Multi-source Spatial Reasoning**
>
> Real environments often include overlapping sound sources, whereas our dataset focuses on single-source localization. BAT[52], which introduced the Spatial AST model, used the encoder for multi-source spatial reasoning. Since our model also adopts the same encoder, we expect it could handle multi-source settings if such examples are included during training. Due to time constraints, we only conducted a zero-shot evaluation, which showed some degradation but generally reasonable behavior. To investigate this further, we conducted zero-shot evaluations under three different multi-source scenarios:
>
>     1. Adding a second sound source with a different sound class at a different location from the original source. (v,v)
>     2. Duplicating an original sound source and putting the copy at a different location from the original. (x,v)
>     3. Adding a second sound source with a different sound class at the same location as the original source. (v,x)
>
> | Different Sound? | Different Location? | Q1 Sound Acc. | Q1 Coming-from Acc. | Q2 Yes/No Acc. | Q3 3-field Acc. | Q3 Dist. Err. (m) | Q4 DoA Acc. | Q4 DoA Err. (°) | Q4 Dist. Err. (m) | Q2-inv. Yes/No Acc. | Q3-inv. 3-field Acc. | Q3-inv. Dist. Err. (m) | Q4-inv. DoA Acc. | Q4-inv. DoA Err. (°) | Q4-inv. Dist. Err. (m) |
> |------------------|----------------------|----------------|----------------------|----------------|------------------|--------------------|--------------|------------------|--------------------|------------------------|--------------------------|--------------------------|--------------------|--------------------------|--------------------------|
> | ✗                | ✗                    | 52.69          | 69.64                | 84.74          | 69.73            | 0.39               | 65.68        | 15.41            | 0.38               | 72.46                  | 59.52                    | 0.75                     | 41.18              | 39.81                     | 0.71                     |
> | ✓                | ✓                    | 31.56          | 59.53                | 83.01          | 70.17            | 0.39               | 63.70        | 17.43            | 0.41               | 71.43                  | 56.17                    | 0.89                     | 35.14              | 48.82                     | 0.78                     |
> | ✗                | ✓                    | 53.75          | 65.39                | 83.66          | 69.42            | 0.42               | 63.20        | 17.78            | 0.41               | 72.46                  | 57.32                    | 0.86                     | 35.18              | 51.04                     | 0.84                     |
> | ✓                | ✗                    | 33.08          | 63.92                | 83.87          | 69.64            | 0.40               | 65.50        | 16.51            | 0.40               | 72.11                  | 58.13                    | 0.81                     | 40.52              | 40.53                     | 0.73                     |
>
> | Different Sound? | Different Location? | Q5 Class Acc. | Q5 Sound Acc. | Q6 Class Acc. | Q6 Sound Acc. | Q7 Class Acc. | Q7 DoA Acc. | Q7 DoA Err. (°) | Q7 Dist. Err. (m) | Q8 Class Acc. | Q8 DoA Acc. | Q8 DoA Err. (°) | Q8 Dist. Err. (m) | Q9 Sound Acc. | Q9 Vis. Acc. |
> |------------------|----------------------|----------------|----------------|----------------|----------------|----------------|--------------|------------------|--------------------|----------------|--------------|------------------|--------------------|----------------|---------------|
> | ✗                | ✗                    | 72.43          | 75.60          | 81.06          | 72.33          | 75.44          | 73.21        | 14.75            | 0.30               | 70.27          | 64.27        | 23.78            | 0.36               | 54.00          | 75.22         |
> | ✓                | ✓                    | 72.82          | 71.21          | 78.72          | 61.39          | 61.71          | 64.63        | 25.15            | 0.39               | 58.09          | 51.74        | 39.46            | 0.43               | 32.42          | 65.42         |
> | ✗                | ✓                    | 72.10          | 72.43          | 80.89          | 58.83          | 74.78          | 70.30        | 17.66            | 0.32               | 71.96          | 52.69        | 39.36            | 0.41               | 54.56          | 75.89         |
> | ✓                | ✗                    | 72.21          | 74.37          | 79.50          | 71.39          | 65.42          | 69.04        | 19.23            | 0.37               | 59.04          | 62.88        | 25.17            | 0.39               | 33.83          | 65.75         |
>
>
> Among these results, we would like to highlight the metrics that show significant differences across experimental settings.
>
> Classification accuracies (e.g., Q1, Q7, Q8, Q9): These results are relatively trivial and do not reveal substantial variation.
>
> *Q1 – Coming-from Accuracy:*
>
> The “Same Sound, Different Location (x,v)” setting shows the highest performance. This is because the model does not fail in sound classification, and the spatial alignment between the original sound source and the corresponding visual object is preserved without contamination. The better performance of the “Different Sound, Same Location (v,x)” setting compared to the “Different Sound, Different Location (v,v)” setting is also reasonable. Even when classification is incorrect, the model can still localize the sound effectively, and the visual encoder can correctly recognize the object at that location.
>
> *Q4 – DoA Accuracy:*
>
> DoA performance unsurprisingly degrades under “Different Location” settings, (x,v), (v,v). The performance gap becomes larger when the visual modality is unavailable. This highlights the importance of visual input in supporting spatial localization.
>
> *Q6 – Sounding Accuracy:*
>
> This metric is relatively straightforward. The question requires spatial audio understanding to identify which visual object is producing the sound. “Different location (x,v), (v,v)” introduces interference, and the presence of a second sound from the same category (x,v) adds further confusion.
>
> *Q7 vs. Q8*:
> In Q7, the settings with both “Same sound, Different location” or “Different sound, same location” outperform the “Different sound, different location” setting. In contrast, in Q8, the “Same sound, Different location” condition performs worse, even though it achieves the highest classification accuracy. This discrepancy stems from differences in scene composition.
> Q7 contains only one visual object from the target category. Therefore, even with two audio sources from the same category, the level of confusion remains limited.
> Q8 includes two visual objects from the same category and two audio sources from the same category, which increases the complexity of spatial reasoning and makes disambiguation more difficult.
>
> *Q9 – Visibility Accuracy:*
> As long as the model correctly classifies the object, visibility accuracy remains unaffected.
>
> We are grateful for the insightful suggestion, which significantly improved the quality of our work.
>
>
> **Stitching Artifacts**
>
>
> While we could not conduct the blurring experiment within the rebuttal period, we respectfully refer you to Figures 4 and 5 of the supplementary material. These examples illustrate that spatial audio is still required for accurate localization, even in the presence of stitching artifacts. In both scenes, the candidate objects (e.g., two chickens or two electric blenders) belong to the same visual category and exhibit similar stitching cues (though these are not particularly salient to the human eye), making it difficult for the model to rely on low-level visual artifacts alone. This suggests that the model needs spatial audio cues to correctly disambiguate the true source. Additionally, the cat next to the chicken on the right side appears to have similar stitching cues, and semantic understanding is still required to distinguish the cat from the chicken despite the presence of such visual seams.

---

> ### Comment · Reviewer_doKX · 2025-08-05
>
> Thank you for addressing most of my concerns and including the additional experiments. The multi-source experiment is a great demonstration. Please make sure to include it in the revised version.
>
> After reading the rebuttals, I would like to keep my original rating. What I really wish is that you could provide real-world evaluations, which I think are crucial for evaluating generalization. You may consider evaluating in-the-wild datasets like StereoWalks [1] in the future.
>
> [1] Min et al. Supervising Sound Localization by In-the-wild Egomotion. CVPR 2025.

---

> > ### Author Response · Authors · 2025-08-05
> >
> > Thank you very much for your thoughtful feedback. We are glad to hear that your concerns have been **mostly addressed.** We also appreciate your positive comment on the multi-source experiment, and are grateful that you found it to be a **great demonstration.** We will make sure to include it in the revised version. Unfortunately, StereoWalks [1] is **not publicly available yet**. However, we agree that the paper can serve as a basis for real-world evaluation in the future. We sincerely appreciate your efforts throughout the review process.

---

### Official Review · Reviewer_TUUu · 2025-07-02

**Clarity:** 2
**Significance:** 3
**Originality:** 2
**Rating:** 3
**Confidence:** 4

**Summary:**

This paper introduces a dataset of spatial audio-visual question answering, which targets to evaluate the model capability of reasoning about sound origins (particularly the location) given the accompanying visual perception. The authors also propose a promising model to address this task, which generally outperforms the baselines. Additional ablation studies reveal the nature of each type of question.

**Questions:**

Q1. I didn't see a discussion that the visual and audio perception may have position mismatch because of the physical volume of the sensors, e.g., in many robots, the audio sensor is usually below the visual one. Have you taken this into consideration in design?

If possible, it would also be good to include the ground-truth positions of these sensors in the dataset.

Q2. Did you also finetune the baseline models in Table 2?

**Ethical Concerns:**

["NO or VERY MINOR ethics concerns only"]

**Final Justification:**

(not yet finalized, pending discussion with fellow reviewers and ACs)

On the one hand, I appreciate the authors for adding so many new experiments; on the other hand, the new stuff lacks rigorous peer review. Additionally, I feel it's bizarre to have a comic as Figure 1---we are doing rigorous science---but the authors didn't respond regarding this.

I am currently hesitating to accept this paper.

**Limitations:**

Yes.

**Quality:**

2

**Strengths And Weaknesses:**

In what follows, I will use S, W, C, and Q to denote Strengths, Weaknesses, Comments, and Questions, respectively, for ease of reference in the discussion.

S1. The task is highly interesting and important in embodied AI.

S2. The presented model is fundamentally simple, and the authors have included a comprehensive analysis through ablation studies for all covered categories.

W1. I am concerned about the QA setup. While it provides a natural language interface to understand the model performance, the current QA setup has too many confounding factors---for example, a poor performance could be because of (1) the model's inability to understand the question, (2) the model's inability to understand the visual input, (3) the model's inability to understand the audio input, (4) the model's inability to reason about the relationship between the audio and visual inputs, or (5) the model's inability to translate the reasoning into a natural language answer, to name a few inexhaustive potential reasons. I don't see a reason why not release the standard annotations and make the QA interface a higher-level interface on top of the standard annotations, which would make the evaluation more interpretable and easier to understand, as well as enabling more systematic analyses of the model performance.

I understand that the authors may be affected much by the "standard" approaches of the current community in building benchmarks, among which many simply release a bunch of QA pairs for anything, but I would strongly encourage the authors to consider more principled approaches when conducting scientific research.

There could also have been much more interesting ablation studies and analyses with more structured annotations, for example, how does slightly moving an object (represented by different input scenarios) affect the model reasoning? Is there an internal neuron that accounts for a specific type of task?

While I'm not suggesting the authors conduct all the suggested experiments above in one paper, I feel the impact of the current data format will be highly limited, and I suggest the authors consider more controlled environments that enable systematic analyses of the model performance. A good example for what to consider and what to release in a dataset, though a bit out of domain, is https://openreview.net/forum?id=84pDoCD4lH

W2. There is an important missing reference: when it comes to sound localization, this paper https://openaccess.thecvf.com/content_ECCV_2018/papers/Relja_Arandjelovic_Objects_that_Sound_ECCV_2018_paper.pdf comes to my mind but is unfortunately not cited in this paper. I would suggest the authors do a more comprehensive literature review.

W3. While I understand this is unfortunately not a standard process in AI/ML, the paper could benefit a lot from reporting the standard errors in the figures and tables---for example, it wouldn't be hard to add a standard error to the results in Tables 2 and 3, and this would add another theoretical layer of confidence to the findings.

W4. There are a few technical designs that I would suggest the authors consider revising:

1. The visual encoder is SigLIP, a CLIP-style image and text matching model, which may not sufficiently elicit the visual features that are particularly important for this task. I would suggest the authors consider using Dino as an additional source of visual features.

C1. This doesn't affect my judgment on the technical contribution of the paper, but I strongly object to having a comic teaser in a scientific paper. It would be good to replace Figure 1 with a more scientifically rigorous figure that better illustrates the point, and it would be a good usage of the limited spaces of the main content. Such content should better go to social media or other channels for dissemination, not in the main paper nor an academic poster presentation.

C2. I suggest some more direct illustrations through figures, instead of a huge table for the ablation study in Table 3. When using tables, it's also good to align the decimal points for better readability---the current presentation is quite hard to digest.

---

> ### Author Rebuttal · Authors · 2025-07-30
>
> We sincerely appreciate your suggestion to analyze the question answering setup. This encouraged us to closely examine our framework and provide a clearer breakdown of its components.
>
> **W1**
>
> We acknowledge your concern about potential confounding factors in QA-based evaluation, such as failures stemming from misinterpreting inputs or cross-modal reasoning.
>
> To address these concerns, we provide two diagnostic tables.
>
> Modality Understanding:
>
> The first table demonstrates that the model is able to process each modality independently and align them within the same latent space as the LLM, ensuring that the encoders generate meaningful and well-structured representations.
>
> As we mentioned in the paper, to ensure the effectiveness of each modality-specific representation, the audio and visual encoders, along with their respective projectors, are pretrained in a unimodal setting using a large language model. For details, please refer to line 208 in the paper.
>
> ***Performance metrics of Audio and Vision modalities***
> | Modality | Detection | avg. Angular Error (°) | Angular Error > 30 | avg. Distance Error (m) |
> |----------|-----------|------------------------|---------------------|--------------------------|
> | Audio    | 0.575     | 38.01                  | 28.9%               | 0.476                    |
> | Vision   | 0.633     | 26.89                  | 16.1%               | 0.332                    |
>
>
> After the warm start of the encoders, each modality encoder performs reasonably well on detection and localization within its own modality.
>
> LLM Reasoning:
> The second table evaluates the LLM’s ability to answer questions when given perfect modality representations. This isolates its reasoning and language capabilities from perception-related errors. We follow Oracle experiment of BAT[52].
>
> All metrics, except the 3-field classification in Q3, the DoA accuracy in Q4, and the DoA accuracy in Q7,8, show almost perfect results. The relatively lower performance on Q3 and Q4 seems to stem from the limited mathematical ability of the LLM we used, as these questions require calculating spatial relationships between two objects and the receiver. In Q7,8, the DoA accuracy is affected by the presence of a non-sounding object that belongs to the same semantic category as the sounding object in prompt.
>
> ***Oracle performance (O+Q)***
> | Metric       | Q1 Sound Acc. | Q1 Coming-from Acc. | Q2 Yes/No Acc. | Q3 3-field Acc. | Q3 Dist. Err. (m) | Q4 DoA Acc. | Q4 DoA Err. (°) | Q4 Dist. Err. (m) | Q2-inv. Acc. | Q3-inv. 3-field Acc. | Q3-inv. Dist. Err. (m) | Q4-inv. DoA Acc. | Q4-inv. DoA Err. (°) | Q4-inv. Dist. Err. (m) |
> |--------------|----------------|----------------------|----------------|------------------|--------------------|--------------|------------------|--------------------|---------------|------------------------|--------------------------|--------------------|--------------------------|--------------------------|
> | Oracle (O+Q) | 98.08          | 95.61                | 94.70          | 85.21            | 0.11               | 86.63        | 4.17             | 0.16               | 94.63         | 84.31                  | 0.10                     | 86.24              | 4.27                     | 0.15                     |
>
> | Metric       | Q5 Class Acc. | Q5 Sounding Acc. | Q6 Class Acc. | Q6 Sounding Acc. | Q7 Class Acc. | Q7 DoA Acc. | Q7 DoA Err. (°) | Q7 Dist. Err. (m) | Q8 Class Acc. | Q8 DoA Acc. | Q8 DoA Err. (°) | Q8 Dist. Err. (m) | Q9 Sound Acc. | Q9 Vis. Acc. |
> |--------------|----------------|-------------------|----------------|-------------------|----------------|--------------|------------------|--------------------|----------------|--------------|------------------|--------------------|----------------|---------------|
> | Oracle (O+Q) | 97.50          | 100               | 95.78          | 100               | 93.83          | 92.05        | 4.05             | 0.11               | 95.15          | 90.67        | 3.86             | 0.12               | 98.03          | 100           |
>
>
> These analyses help validate whether the modality understanding and question-answering components function as intended. Most of the time, reasoning is not the bottleneck. Errors tend to arise either from perception or from failures in aligning modality-specific representations with the LLM’s reasoning capabilities.
>
> Large language models and question answering datasets allows new question types or answer categories to be added with minimal cost, often without the need to retrain the entire model. In contrast, classifier-based approaches require predefined categories and typically demand full retraining to accommodate new types. By leveraging the generative nature of language, we can explore broader and more realistic evaluation scenarios beyond the constraints of closed-form classification.
>
> We will incorporate the aforementioned experiments into the paper to encourage more systematic analysis of each component involved in multimodal question answering.
>
> **W2**
>
> We appreciate the suggestion to include this relevant work. In our paper, we discuss an earlier pioneer work on sound localization [37], as well as several more recent approaches [21,7,23,42,29,39,43]. Nonetheless, we will include this reference in the revised version for completeness.
>
>
> **W3**
>
> We agree that reporting standard errors adds confidence to the findings. Due to limited time and GPU resources, we repeated the two most important experiments, R+B+Q and R+M+Q. As shown, the performance differences are minimal. However, we would like to highlight that metrics which rely on spatial audio consistently show clear differences between the two settings across trials (e.g., Q3-inv 3-field, Q4-inv DoA, Q6-sound, Q8-DoA).
>
> | Metric       | Q1 Sound Acc. | Q1 Coming-from Acc. | Q2 Yes/No Acc. | Q3 3-field Acc. | Q3 Dist. Err. (m) | Q4 DoA Acc. | Q4 Err. (°) | Q4 Dist. Err. (m) | Q2-inv. Acc. | Q3-inv. 3-field Acc. | Q3-inv. Dist. Err. (m) | Q4-inv. DoA Acc. | Q4-inv. DoA Err. (°) | Q4-inv. Dist. Err. (m) |
> |--------------|----------------|----------------------|----------------|------------------|--------------------|--------------|--------------|--------------------|---------------|---------------|--------------------------|---------------|--------------------|--------------------------|
> | R+B+Q (1)    | 52.69          | 69.64                | 84.74          | 69.73            | 0.39               | 65.68        | 15.41        | 0.38               | 72.46         | 59.52         | 0.75                     | 41.18         | 39.81              | 0.71                     |
> | R+B+Q (2)    | 51.61          | 70.92                | 82.68          | 69.10            | 0.35               | 65.14        | 15.14        | 0.34               | 72.91         | 53.75         | 0.76                     | 42.59         | 37.91              | 0.69                     |
> | R+M+Q (1)    | 51.01          | 64.10                | 83.77          | 66.52            | 0.41               | 59.01        | 20.21        | 0.47               | 70.40         | 47.29         | 0.98                     | 69.25         | 16.71              | 1.08                     |
> | R+M+Q (2)    | 47.39          | 65.06                | 83.33          | 67.70            | 0.37               | 61.88        | 18.69        | 0.49               | 69.37         | 41.98         | 1.06                     | 71.12         | 15.65              | 1.08                     |
>
> | Metric       | Q5 Class Acc. | Q5 Sound Acc. | Q6 Class Acc. | Q6 Sound Acc. | Q7 Class Acc. | Q7 DoA Acc. | Q7 Err. (°) | Q7 Dist. Err. (m) | Q8 Class Acc. | Q8 DoA Acc. | Q8 Err. (°) | Q8 Dist. Err. (m) | Q9 Sound Acc. | Q9 Vis. Acc. |
> |--------------|----------------|----------------|----------------|----------------|----------------|--------------|--------------|--------------------|----------------|--------------|--------------|--------------------|----------------|---------------|
> | R+B+Q (1)    | 72.43          | 75.60          | 81.06          | 72.33          | 75.44          | 73.21        | 23.78        | 0.36               | 70.27          | 64.27        | 23.78        | 0.36               | 54.00          | 75.22         |
> | R+B+Q (2)    | 74.87          | 72.98          | 81.21          | 66.53          | 75.02          | 71.64        | 24.32        | 0.34               | 74.89          | 63.39        | 29.38        | 0.34               | 51.63          | 73.69         |
> | R+M+Q (1)    | 74.26          | 64.54          | 81.61          | 53.33          | 75.33          | 67.40        | 30.47        | 0.44               | 75.33          | 57.84        | 30.47        | 0.44               | 51.14          | 72.94         |
> | R+M+Q (2)    | 78.15          | 65.15          | 83.94          | 51.11          | 75.22          | 68.53        | 32.01        | 0.45               | 78.39          | 46.22        | 29.63        | 0.45               | 47.31          | 66.31         |
>
>
> **W4**
>
> We use SigLIP2 with the NaFLEX setting (Line 225), which allows flexible image resolutions and aspect ratios. This is crucial for our use of 360-degree panoramic images, which have a 4:1 width-to-height ratio. DINO assume square inputs, which distort our panoramic images. While we agree that DINO offers strong features, it is not compatible with our setting. SigLIP2 provides a better fit due to its geometric flexibility.
>
> **Q1 Answer**
>
> Please read rebuttal to yWdi, ***From the reviewers***
>
> **Q2 Answer**
>
> We finetuned VideoLLaMA2 on our dataset, but did not finetune ISSL or ACL-SSL. They are originally trained on datasets which is source of audio clip of our dataset (VGGSound). Additionally, since ISSL and ACL-SSL use different visual encoders (ResNet-18 and CLIP, respectively), a perfectly fair comparison with our model (which uses SigLIP2) is not feasible. More details are provided in Section A.1.1 of the supplementary material.

---

> > ### Comment · Reviewer_TUUu · 2025-08-06
> > **Thanks for your response.**
> >
> > Thanks for the additional results and your careful response. While I would tend to recommend another round of review for the newly added results (in addition to improving the scientific rigor of this paper, which are presented in my comments but have not received authors' response), I believe the current results form a much stronger story than the initial shape. I'm still on the conservative side, but would be happy to discuss with other reviewers regarding the paper. Now I'll put my rating to a 3.

---

> > > ### Author Response · Authors · 2025-08-06
> > >
> > > Thank you very much for your thoughtful reconsideration and for kindly updating your rating. We truly appreciate your openness to further discussion.
> > >
> > > Regarding the point on **scientific rigor**, we would like to clarify that **we do have standard annotations** aligned with the QA format, as our QA pairs are verbalized forms of these annotations. This makes the task applicable to models that do not rely on LLMs. For example, our standard annotations could be used in the same manner as Stage 1 of BAT [52], which uses classifier-based supervision to pretrain the Spatial-AST model. **We promise that we will release the standard annotations to enable more rigorous and systematic evaluation.**
> > >
> > > If helpful, we would like to respectfully refer to some of the responses we shared with other reviewers during the rebuttal phase, which may also contribute to your better understanding.
> > >
> > > In particular, we provided an analysis of performance regarding the "Comparison with stronger LLMs" (in our response to Reviewer WvgB), which we believe complements our analysis in W1 by exploring the upper bounds of reasoning performance. **We sincerely appreciate that this analysis was inspired by your suggestion in W1**.
> > >
> > > We also conducted a multi-source evaluation (as described in our response to Reviewer doKX), which doKX described as a **great demonstration.** We believe this analysis helps clarify the contribution of each modality and supports a more interpretable evaluation framework.
> > >
> > > We are truly grateful for your thoughtful feedback throughout the process, which have significantly helped us refine the scope and clarity of the paper.

---

### Official Review · Reviewer_WvgB · 2025-07-03

**Clarity:** 3
**Significance:** 3
**Originality:** 2
**Rating:** 4
**Confidence:** 4

**Summary:**

This paper proposes a new task audio-visual spatial reasoning, which requires the language model to perform both spatial localization and semantic matching for audio-visual scene. To address this, the authors introduce a new large-scale simulated dataset, consisting of 1 million qa pairs for understanding complex spatial relationships between sound and visual context in 3D environments. The authors also propose a multi-modal llm to integrate spatial audio and visual encoders with a language backbone to perform these tasks. Empirical evaluations demonstrate that the approach outperforms strong audio-visual baselines, supporting the core claims that incorporating binaural audio enables richer spatial reasoning.

**Questions:**

When preparing the rebuttal, I would suggest that the authors focus on the main weaknesses as outlined above. To summarize:

1. Real-World Gap: can the authors comment more concretely on how they foresee bridging the simulation-to-real gap?
2. More comparison methods: including gemini and qwen-omni, at least on part of the dataset.
3. Clarify how the model produces such fine spatial coordinates from binaural audio alone.

**Ethical Concerns:**

["NO or VERY MINOR ethics concerns only"]

**Limitations:**

Yes, the authors adequately addressed the limitations and potential negative impact of their work.

**Quality:**

2

**Strengths And Weaknesses:**

Strengths:

1. The proposed “Hear You Are QA” dataset covers a wide range of spatial reasoning problems, including alignment/disambiguation tasks that prior datasets and benchmarks did not address. The question types provided in tab1 give a clear taxonomy, motivating why audio-visual spatial reasoning matters.

2. The authors also propose a model, adopting modality-specific encoders and projectors to transform vision and audio data into representations to be fed to the large language model. The model is capable of doing spatial audio-visual understanding, which is new.
Since there isn’t spatial audio-visual llm baselines to be compared with, the authors made great efforts to adapt existing non-llm spatial audio-visual methods and videollama2 for comparison.


Weaknesses:

1. The model relies on a Q-Former-style projector to compress vision and audio features before passing them to the LLM. This type of compression often loses information, which is why many recent works prefer the simpler LLaVA-style MLP projector. Why did the authors keep the Q-Former method rather than switching to a lightweight MLP? A clearer justification is needed.

2. The whole dataset is rendered with SoundSpaces. Both the images, using meshes, and the binaural audio, using a fixed HRTF, are synthetic. How will the system transfer to real-world RGB frames and real binaural recordings? The paper didn’t show real-world test results, so the generalization ability is still unknown.

3. The model predicts very precise coordinates such as (−30, −12), 2.549m from binaural audio only. In practice, such accuracy usually depends on the carefully designed microphone array and actual scene geometry visible in the image. Are these outputs absolute or relative to the rendered environment? I would like the authors to clarify how this would work on real data.

4. There are multimodal LLMs already able to reason about spatial audio, e.g., Gemini 2.5-Flash/Pro. Qwen-Omni is also known to handle audio-visual inputs and should outperform VideoLLaMA-2. The authors should run or at least discuss these stronger baselines.

---

> ### Author Rebuttal · Authors · 2025-07-30
>
> We truly appreciate your thoughtful questions on the real-world scenario, the precision of spatial localization from binaural audio alone, comparisons with stronger LLMs, and the projector design. In particular, your question on stronger LLMs inspired us to explore future opportunities.
>
>
> **Real-world Scenario Discussion**
>
> Please read rebuttal to yWdi
>
> **How the model produces such fine spatial coordinates from binaural audio alone?**
>
> We apologize for the confusion. Our model does not achieve such high precision; in our case, both training and evaluation are performed with 0.1-meter precision. As we are building a benchmark, we annotated the dataset with coordinates up to three decimal places to support more precise localization in future models.
>
> **Comparison with stronger LLMs**
>
> Regarding the stronger baseline, we appreciate your suggestion. While we were unable to implement the Qwen-Omni-based Hear You Are LLM due to time constraints, we appreciate your understanding. Nevertheless, we would like to outline the expected implementation of a Qwen-Omni-based Hear You Are LLM and discuss its feasibility.
>
> First, Qwen2.5-Omni explicitly supports only mono audio, as stated in its official GitHub repository:
>
> qwen-omni-utils/src/qwen_omni_utils/v2_5/audio_process.py, Line 50:
> raise ValueError("Support only mono audio")
>
> In addition, its ViT-based vision architecture does not explicitly support the 4:1 image aspect ratio required for our panoramic image input. Therefore, to make this implementation compatible with our setup, we would need to replace both the audio encoder (e.g., with Spatial-AST) and the vision encoder (e.g., with SigLIP2 NaFLEX).
>
> However, with these two major components substituted, the distinctive element remaining from the original baseline would be the large language model (LLM). Accordingly, we instead chose to provide a focused discussion on the potential performance gains from using a stronger LLM.
>
> We would like to refer to the oracle performance provided in the rebuttal to TUUu, where Q3 and Q4, which require mathematical reasoning, show relatively low accuracy. We hypothesize that this is due to the limited mathematical capability of Qwen2-7B.
>
> Notably, Qwen2.5 demonstrates significant improvements in arithmetic tasks:
>
> | Model           | MATH (%) |
> | --------------- | -------- |
> | Qwen2-7B        | 52.9     |
> | Qwen2.5-7B      | 75.5     |
> | Qwen2.5-Omni-7B | 71.5     |
>
> This suggests that using Qwen2.5-Omni’s LLM could raise the upper bound for performance on questions such as Q3 and Q4, which involve numerical and spatial reasoning.
>
> Additionally, to the best of our knowledge, Gemini 2.5-Flash/Pro currently does not support spatial or binaural audio understanding.
>
> Gemini 2.5 Pro does not mention any capability related to spatial reasoning with audio in its technical report. To evaluate whether it possesses spatial understanding, we conducted a simple experiment. We used audio samples from our supplementary material, excluding those labeled as directly in front or directly behind, and prompted Gemini to classify whether the sound was coming from the left or the right. Our empirical testing showed that its performance was no better than random guessing. Moreover, when we repeated the same input three times, we observed inconsistent responses across trials. This suggests a lack of consistent spatial audio understanding.
>
> prompt: Given the binaural audio, answer which direction is the sound is coming from. Please answer left or right.
>
> ***Prediction Comparison Table***
>
> | Filename   | GT    | Pred1 | Pred2 | Pred3 |
> |------------|-------|-------|-------|-------|
> | q1_0.wav   | right | right | right | left  |
> | q2_0.wav   | left  | left  | left  | right |
> | q2_1.wav   | left  | left  | left  | right |
> | q2_2.wav   | left  | right | right | right |
> | q3_0.wav   | right | right | right | left  |
> | q3_2.wav   | right | left  | right | left  |
> | q4_0.wav   | right | right | left  | right |
> | q4_2.wav   | left  | left  | left  | right |
> | q5_0.wav   | right | left  | right | left  |
> | q5_1.wav   | left  | right | right | left  |
> | q5_2.wav   | right | right | left  | left  |
> | q6_0.wav   | right | right | left  | right |
> | q6_1.wav   | left  | left  | right | right |
> | q6_2.wav   | left  | right | right | right |
> | q7_0.wav   | right | left  | right | left  |
> | q7_2.wav   | right | right | left  | left  |
> | q8_1.wav   | right | left  | right | right |
> | q8_2.wav   | right | right | left  | right |
> | q9_0.wav   | left  | left  | right | left  |
> | q9_1.wav   | left  | right | right | left  |
> |  **Accuracy**          |       | 0.60  | 0.45  | 0.35  |
>
>
> As we are introducing a new task, we acknowledge the lack of well-established baselines. Even existing sound source localization models are not directly comparable to our setup. To provide a meaningful point of reference, we fine-tuned VideoLLaMA2 on our dataset and carefully aligned the experimental settings to serve as a fair and transparent baseline. We appreciate your consideration of our efforts to build a fair point of comparison in the absence of existing baselines.
>
> **Design choice of Q-Former**
>
> We appreciate the point. We followed the official Spatial-AST design, which uses a Q-Former for audio, and applied it to SigLIP2 for consistency.
>
> We agree that an MLP is a promising alternative. However, we also want to emphasize that specific architecture of the audio and video encoder themselves is not the main contribution of work. There will be many good choices. We are presenting the dataset and benchmark and functioning model, which is non trivial. There will be a lot of opportunity of future work to improve.

---

### Official Review · Reviewer_yWdi · 2025-07-07

**Clarity:** 3
**Significance:** 3
**Originality:** 3
**Rating:** 5
**Confidence:** 4

**Summary:**

The authors address the problem of imparting spatial comprehension of relative locations in space of sound sources using vision and binaural audio. The authors propose a new entirely synthetic dataset constructed using SoundScape 2.0, for generating audio and Matterport3D + Stable diffusion for creating the visual scene with the corresponding visual objects representing source of sound or distractors. They use this dataset to train their model - "Hear you are" LLM which essentially comprises of 1. Modality specific encoders:  visual encoder to tokenize the panoramic view of the visual scene and a binaural audio encoder to encode the spatial audio and a text encoder. 2. Modality specific Q-Formers 3. An LLM with LoRA adapters. As this is a novel task, there are no existing benchmarks and baseline models. So the authors resort to creating some interesting baselines: 2 models based on audio-Visual sound source localization and another baseline with the exact configuration as the proposed approach but with mono audio (lacking spatial information of sound sources). Since the dataset is synthesized it allows for crafting Questions and their respective Answer. Authors construct 9 categories of questions to span questions probing the model about relative distances, semantic categorization of sound sources, disambiguating sounds from plausible sources in the visual scene, and localizing source when it is visible and not visible in the visual scene.

**Questions:**

See the Strengths and Weakness section.

**Ethical Concerns:**

["NO or VERY MINOR ethics concerns only"]

**Quality:**

3

**Strengths And Weaknesses:**

Strengths
1. The paper is very well written, easy to understand and widely accessible.
2. The proposed task and dataset are relevant and the evaluations are very interesting. Clearly shows the benefit of video and binaural audio. This is a good initial work in this area.

Weakness:
1. Entire dataset is synthetic and the real challenge is in making this work in the real world. It would be good to supplement a large syntetic data with some real data to assess feasibility of generalization. However this in itself would be a pretty significant task to design real world data collection and having the synthetic environments mimic that.

---

> ### Author Rebuttal · Authors · 2025-07-30
>
> We truly appreciate your positive assessment of our work, especially acknowledging the novelty and clarity of our proposed task and dataset. While our study is not without limitations, we are grateful that you recognized the challenge and complexity of real-world data collection.
>
> In response to the common concern regarding real-world generalization, we have compiled a structured discussion. It’s not a complete solution, but we hope it shows that we have seriously considered the key issues and are thinking critically about how to bridge the sim-to-real gap. We hope this structured discussion is helpful in addressing the concern.
>
> **Regarding Real-World Applicability**
>
> We fully agree with the importance of evaluating real-world generalization. Given the time and resource constraints, we were unable to conduct real-world experiments during this rebuttal.
>
> As yWdi noted, “itself would be a pretty significant task to design real world data collection and having the synthetic environments mimic that.” We agree that real-world data collection is undeniably costly and complex. Nevertheless, we believe it is essential for future work. A small-scale evaluation set could serve as a starting point to assess how well the model generalizes from simulation to reality (sim2real). Depending on the size of the gap, this would help determine how much real-world training data would be needed for fine-tuning.
>
> As seen in prior work such as BAT[52], defining new simulation scenarios that challenge existing models and evaluating their capabilities in those settings can serve as meaningful contributions. In particular, it allows researchers to demonstrate the feasibility of applying a model to a new task before making a heavy investment in dataset collection. Our work follows this philosophy and aims to encourage future research in spatial audio-visual understanding.
>
> Instead, we would like to respond to Reviewer WvgB’s request “can the authors comment more concretely on how they foresee bridging the simulation-to-real gap?” by providing a structured discussion on what it would take to achieve this, incorporating insights from existing literature, and reviewer suggestions.
>
> ***From the literature***
>
> We highlight two notable efforts. SONICVERSE[1*] introduced three practical strategies: (1) mixing ego-noise from real robot recordings during training, (2) varying source gain to address loudness mismatch, and (3) calibrating depth sensor range to match real-world hardware. Chen et al. [2*] addressed the issue that ray-tracing-based acoustic simulators degrade at low frequencies, and proposed a frequency-adaptive strategy that selects optimal sub-bands based on spectral energy and prior error trends, improving sim-to-real performance in acoustic field prediction.
>
> [1*] Gao et al., Sonicverse: A Multisensory Simulation Platform for Embodied Household Agents that See and Hear, ICRA 2023.\
> [2*] Chen et al., Sim2Real Transfer for Audio-Visual Navigation with Frequency-Adaptive Acoustic Field Prediction, IROS 2024.
>
> ***From the reviewers***
>
> TUUu noted that real robots often have physical offsets between visual and audio sensors, such as a microphone located below a camera. While our setup assumes co-located sensors for simplicity, we agree that such offsets can be incorporated during simulation. Moreover, from the perspective of the object rather than the sensor, applying augmentations such as assigning sound sources to varied positions within the object volume could help improve spatial generalization.
>
> doKX pointed out that real environments often include overlapping sound sources, whereas our dataset focuses on single-source localization. BAT, which introduced the Spatial AST model, used the encoder for multi-source spatial reasoning. Since our model also adopts the same encoder, we expect it could handle multi-source settings if such examples are included during training. Due to time constraints, we only conducted a zero-shot evaluation (see Table of the rebuttal to doKX), which showed some degradation but generally reasonable behavior. We would appreciate the reviewers’ understanding that our current focus is on ‘exploring audio-visual spatial alignment beyond semantic alignment’, and we consider multi-source learning as an important direction for future work.
>
> In summary, while our current study focuses on audio-visual spatial alignment in a controlled setting, we view it as a first step toward audio-visual understanding in the real world. We appreciate the reviewers’ insightful comments, which have helped us identify important future directions.

---

### Note · Authors · 2025-08-14

Final Remarks

We sincerely thank you for your thoughtful reconsideration and for updating your rating.

Our dataset is inherently based on standard annotations fully aligned with the QA format, as the QA pairs are verbalized forms of these annotations. This ensures the task is applicable to models without relying on LLMs. For example, the standard annotations could be used similarly to Stage 1 of BAT [52], which employs classifier-based supervision to pretrain the Spatial-AST model. We will release these annotations to support more rigorous evaluation.

Due to the 6000-character limit during the first rebuttal, this clarification on standard annotations could not be addressed at that stage. We have included it in the official comment following the reviewer feedback, and we would like to kindly draw your attention to it, as we believe it is essential and fully addresses the reviewers’ concerns. We are optimistic that this clarification will help the discussion converge toward a positive consensus.

Best regards,\
Authors

---

### Decision · Program_Chairs · 2025-09-17

**Decision:**

Reject

**Comment:**

This paper introduces a dataset of spatial audio-visual question answering and proposes a model for the task. Most reviewers gave positive ratings and pointed out the strengths, mainly including:
- Interesting and important topic
- The proposed dataset covers a wide range of spatial reasoning problems.

Some reviewers also raised some concerns (including some reviewers giving positive recommendations), such as lack of real-world evaluation, unsatisfactory writing, missing baselines and experiments, etc.

After reading the paper carefully, I admitted that it has some merits, but I partially agreed with the concerns raised by Reviewer TUUu. Except for the new dataset, the remaining part is negligible. Especially, how the visual-audio spatial reasoning mechanism is realized is quite vague: it simply relies on the LLM to realize the magic and I don't believe that LLM can do this correctly as the underlying principle is not elaborated at all.

In sum, I'm inclined to reject it and encourage the authors to submit their paper to other conferences and journals, especially benchmark track.